# The Value of Local Therapies in Advanced Adrenocortical Carcinoma

**DOI:** 10.3390/cancers16040706

**Published:** 2024-02-07

**Authors:** Otilia Kimpel, Barbara Altieri, Marta Laganà, Thomas J. Vogl, Hamzah Adwan, Tina Dusek, Vittoria Basile, James Pittaway, Ulrich Dischinger, Marcus Quinkler, Matthias Kroiss, Soraya Puglisi, Deborah Cosentini, Ralph Kickuth, Darko Kastelan, Martin Fassnacht

**Affiliations:** 1Division of Endocrinology and Diabetes, Department of Medicine, University Hospital, University of Würzburg, 97070 Würzburg, Germany; altieri_b@ukw.de (B.A.); dischinger_u@ukw.de (U.D.); matthias.kroiss@med.uni-muenchen.de (M.K.); fassnacht_m@ukw.de (M.F.); 2Medical Oncology Unit, Department of Medical & Surgical Specialties, Radiological Sciences & Public Health, University of Brescia, ASST Spedali Civili of Brescia, 25123 Brescia, Italy; martagana@gmail.com (M.L.); deborah.cosentini@gmail.com (D.C.); 3Universitätsklinikum Frankfurt, Institut für Diagnostische und Interventionelle Radiologie, 60596 Frankfurt, Germany; t.vogl@em.uni-frankfurt.de (T.J.V.);; 4Department of Endocrinology, University Hospital Centre Zagreb, School of Medicine, University of Zagreb, 10000 Zagreb, Croatia; tdusek@mef.hr (T.D.); darko.kastelan@kbc-zagreb.hr (D.K.); 5Internal Medicine 1, San Luigi Gonzaga Hospital, Department of Clinical and Biological Sciences, University of Turin, 10043 Orbassano, Italy; basile_vittoria@libero.it (V.B.); soraya.puglisi@unito.it (S.P.); 6Department of Endocrinology, St Bartholomew’s Hospital, London EC1A 7BE, UK; j.pittaway@qmul.ac.uk; 7Endocrinology in Charlottenburg, 10627 Berlin, Germany; marcusquinkler@t-online.de; 8Department of Medicine IV, University Hospital, LMU Munich, Ziemssenstraße 1, 80336 München, Germany; 9Institute of Diagnostic and Interventional Radiology, University-Hospital of Würzburg, 97080 Würzburg, Germany; kickuth_r@ukw.de; 10Comprehensive Cancer Center Mainfranken, University of Würzburg, 97070 Würzburg, Germany

**Keywords:** adrenal cancer, advanced disease, local therapies, radiofrequency ablation, microwave ablation

## Abstract

**Simple Summary:**

Local therapies (LTs) are suggested by most experts and guidelines for the treatment of advanced ACC. However, there are only a few published studies on LTs, and there are no clear recommendations regarding which patients benefit from which treatments. Therefore, this multicentre cohort study aimed to investigate the outcomes and factors associated with LTs (n = 132) when used as a therapeutic approach in 66 patients with metastatic ACC. These patients were treated with local thermal ablation (LTA, n = 84) therapies, transarterial (chemo)embolisation (TA(C)E, n = 40), and transarterial radioembolisation (TARE, n = 8). In 21% of the treated tumoural lesions, complete remission was achieved. Time to progression of the treated lesion was particularly long in patients treated with LTA (median not yet reached), whereas it was only 8.3 months after TA(C)E and 8.2 months after TARE. Thus, this study provides clear evidence that LTs can be quite efficient in a subgroup of patients with advanced ACC.

**Abstract:**

International guidelines recommend local therapies (LTs) such as local thermal ablation (LTA; radiofrequency, microwave, cryoablation), transarterial (chemo)embolisation (TA(C)E), and transarterial radioembolisation (TARE) as therapeutic options for advanced adrenocortical carcinoma (ACC). However, the evidence for these recommendations is scarce. We retrospectively analysed patients receiving LTs for advanced ACC. Time to progression of the treated lesion (tTTP) was the primary endpoint. The secondary endpoints were best objective response, overall progression-free survival, overall survival, adverse events, and the establishment of predictive factors by multivariate Cox analyses. A total of 132 tumoural lesions in 66 patients were treated with LTA (n = 84), TA(C)E (n = 40), and TARE (n = 8). Complete response was achieved in 27 lesions (20.5%; all of them achieved by LTA), partial response in 27 (20.5%), and stable disease in 38 (28.8%). For the LTA group, the median tTTP was not reached, whereas it was reached 8.3 months after TA(C)E and 8.2 months after TARE (*p* < 0.001). The median time interval from primary diagnosis to LT was >47 months. Fewer than four prior therapies and mitotane plasma levels of >14 mg/L positively influenced the tTTP. In summary, this is one of the largest studies on LTs in advanced ACC, and it demonstrates a very high local disease control rate. Thus, it clearly supports the guideline recommendations for LTs in these patients.

## 1. Introduction

Adrenocortical carcinoma (ACC) is a rare and aggressive form of cancer that originates from the adrenal gland and is known for its poor prognoses [1,2,3,4,5,6]. Patients’ prognoses largely depend on tumour stage, resection status, tumour grading, and hormone excess, all of which were recently combined to form the new prognostic S-GRAS score [7]. In its early stages, and with limited risk factors, some patients can be surgically cured, but as the disease progresses to more advanced stages, the prognosis becomes increasingly unfavourable, with 5-year survival rates dropping to less than 20% for those with metastatic disease [1,2,5,8]. However, some patients survive many years, and cases with complete treatment responses have been reported even in patients presenting with stage IV disease [9,10]. 

There are few limited systemic treatment options. International reviews and guidelines advocate the use of mitotane as a primary treatment for recurrent or metastatic ACC, either alone or in combination with platinum-based chemotherapy [3,11,12,13,14,15,16,17,18,19]. This is particularly recommended when complete surgical removal of the cancer is not feasible. Surgical removal of metastatic lesions is typically only considered as a preferred option when all tumour lesions can be completely removed, and only performed when at least 12 months have passed since the last surgery [20,21]. The effects of targeted therapies and immunotherapeutic approaches in ACC patients have also been investigated, but the results are heterogeneous and, for the most part, modest [22,23,24,25,26,27].

In addition, despite limited evidence, the current guidelines suggest consideration of local therapies (LTs) for advanced ACC [11,12]. These local therapeutic measures may include treatments like radiotherapy (RT), transarterial (chemo)embolisation (TA(C)E), and local thermal ablation (LTA) therapies such as radiofrequency ablation (RFA), microwave ablation (MWA), and cryoablation (CA). The choice of LT methods should be based on factors such as the location of the tumour lesion(s), tumour burden, local expertise, prognostic factors, and patient preference [11,12]. Transarterial radioembolisation (TARE) has been described as effective in some case reports [28]. 

Recently, we reviewed the available evidence for radiotherapy as an LT in advanced ACC [29]. We identified 11 studies using radiotherapy in advanced ACC [10,30,31,32,33,34,35,36,37,38,39,40]. In these 11 studies, which included 200 patients, RT was performed as a treatment for irresectable or not completely resectable tumours/metastases, as a therapy for pain, or for the prevention of complications from metastases [10,30,31,32,33,34,35,36,37,38,39,40]. However, at present, only nine studies and a few case reports involving a total of only 170 patients have been published on different types of LT beyond radiotherapy [28,31,41,42,43,44,45,46,47,48,49,50]. The largest of these studies is from Cazejust et al., including 103 lesions. Additionally, Mauda-Havakuk et al. reported 84 lesions, Roux et al. reported with 50, and Veltri et al. reported with 30 lesions [31,41,43,44]. All of these studies reported complete ablation or at least disease control in most of the treated cases, with few reporting adverse events [28,31,41,42,43,44,45,46,47]. Some of these studies analysed the prognostic factors that influenced the efficacy of the LTs. However, only a few possible factors were identified that could predict response. The number of metastases, the size of the treated lesion, and a longer time from first diagnosis to LT seemed to have positive influences on treatment responses [31,42,43,44,45,46]. Nevertheless, there is still no clear recommendation regarding which patients could benefit from which treatments [11,12]. 

Therefore, this multicentre cohort study aimed to investigate the outcomes and factors associated with LTs for patients with advanced ACC. 

## 2. Subjects and Methods

### Study Population

This cohort study was conducted as part of the ENSAT registry study (www.ensat.org/registry, accessed on 1 June 2021) in ten European reference centres for the ACC (Würzburg, Germany; Berlin, Germany; Munich, Germany; Frankfurt, Germany; London, UK; Eindhoven, The Netherlands; Brescia, Italy; Orbassano, Italy; Milano, Italy; Zagreb, Croatia). The study received ethical approval from the ethics committees or institutional review boards at all participating institutions. All patients who participated in this study provided written informed consent.

Only patients with advanced ACC were included. Advanced ACC was defined as disease state, in which the tumour could not be completely removed by surgery. Patients who underwent LTs (excluding radiotherapy) were considered eligible for inclusion. The study included cases treated between 2000 and 2022, with data follow-up extending until October 2023.

Data collection included various demographic, clinical, and histological parameters sourced from both the ENSAT ACC registry and patients’ medical records. These parameters included factors such as sex, age at diagnosis and start of LT, evidence of hormonal overproduction, ENSAT tumour stage [8], information on local and systemic therapies administered before LT, size and number of the treated tumoural lesions, and specific details regarding the LT procedure. Tumour staging at the time of diagnosis relied on imaging studies and findings from surgical procedures and pathological examinations. The accuracy of histological diagnoses was assured through confirmation by experienced pathologists. Patients were excluded if follow-up information was lacking, or if concomitant systemic anti-tumour treatment was initiated up to 12 weeks before and up to 8 weeks after LT (unless progressive disease was already documented). However, patients already receiving mitotane treatment (initiated more than 12 weeks before LT) were eligible, and their mitotane blood levels were recorded. 

## 3. Details on LTs

We endeavoured to accurately capture the technical details of the different LTs where available. LTs were performed at the interventional radiology departments of each centre. All procedures were performed under local anaesthesia with or without conscious sedation or under general anaesthesia with computed tomography (CT) or ultrasound (US) guidance. In all cases, the patients’ vital signs were continuously monitored during the procedure. Imaging was performed after LTs to assess the therapeutic efficacy and potential adverse events. Patients were usually kept under observation overnight and discharged a few days after the treatment.

We made every effort to collect details such as needle length, modality, duration, and tumour diameter for RFA, temperature for MWA and CA, type of embolic particles or the used drug for TA(C)E, and details on TARE if available. 

RFA was performed with different modalities (monopolar vs. bipolar) and via different access (percutaneous, laparoscopic, or open surgery). Local anaesthetic was injected at the site of the electrode needle/antenna insertion and vital parameters (ECG, blood pressure, arterial oxygen saturation) were monitored by the anaesthetist throughout. The procedures were performed under US or CT guidance. LeVeen electrode needles were predominantly used for the RFA technique. The maximum needle length was 3.5 cm and the maximum duration was 20 min. At the end of RFA treatment, a scan of the thorax or abdomen was acquired to rule out immediate complications. 

CT-guided MWA procedures were performed under general anaesthesia. The MWA system consisted of a generator and a water-cooled antenna. Placement of the microwave antenna within the target lesion was performed under CT guidance by an experienced interventional radiologist. The maximum induced power was 140 Watt and the maximum duration was 6 min. Complete necrosis after MWA was confirmed by dynamic contrast-enhanced CT or MRI and the ablation/tumour ratio was calculated for each patient.

The cryoablation procedure was performed using an argon-/helium-gas-based system. The probes were placed under US or CT guidance. The freezing process was monitored in real time to avoid creating lesions on adjacent tissues. 

For TA(C)E, the puncture of the femoral artery with a 5 French catheter was performed by the Seldinger method under local anaesthesia. A selective angiogram was performed and the feeding arteries, tumour, and vascular anatomy surrounding the tumour were identified. Then, a coaxial super-selective microcatheter was inserted through the 5 French catheter, as close to the tumoural lesion as possible. Once the microcatheter was positioned in the target branch, either a combination of chemotherapy and lipiodol or microspheres/particles with/without lipiodol were slowly injected through the catheter affecting embolization, until blood flow has nearly stopped. The doses were determined by the size and vascularity of the tumour. 

TARE was performed with ^90^Y glass microspheres after angiographic tumour mapping with the use of cone-beam CT and treatment simulation with 150-MBq technetium 99 m macroaggregated albumin. 

### 3.1. Outcome Assessment

Before conducting any analyses, we defined the time to progression of the treated lesion (tTTP) as the most relevant outcome. Each treated lesion was evaluated independently. We based the assessment of objective response (OR) on routine radiological evaluations, in analogy to RECIST 1.1 criteria for the treated lesion. Each LT technique was analysed as a separate group. Due to the limited sample size of certain LTs (CA n = 2, MWA n = 18), we grouped these two procedures together with RFA and referred to these techniques as image-guided local thermal ablation therapies (LTAs). We defined “time to progression of the treated lesion” (tTTP) as the duration from the first day of treatment (LT) until the earliest detection of progression in the treated lesion or the date of the last follow-up with imaging. The lesions were also analysed according to the presumed intention to treat (e.g., curative vs. palliative). Patients without any other tumour lesions (except for those treated with LT) were included in a group with a “potentially curative” approach, whereas patients with multiple tumour lesions were classed as the “palliative approach” group. 

Additionally, we conducted an analysis of overall progression-free survival (oPFS), which considers the progression of all tumour lesions, regardless of whether they were treated with LT. Overall progression-free survival (oPFS) was defined as the time elapsed from the initiation of treatment (LT) until either the progression of any lesion or the last follow-up. Finally, “overall survival” (OS) was determined as the time from the date of the first treatment (LT) to either the date of death or the last follow-up. The results of patients who did not experience progression or death were considered as censored data at the last follow-up.

### 3.2. Documentation of Adverse Events

Medical records were examined to assess adverse events related to the use of LTs. These adverse events were retrospectively evaluated using the Common Terminology Criteria for Adverse Events (CTCAE) version 5.0, which provides standardised criteria for grading the severity of adverse events [51].

### 3.3. Statistical Analysis

For all analyses on survival time, we employed the Kaplan–Meier method, and differences between the groups were evaluated using log–rank statistics. In addition, we conducted univariate analyses to explore factors that could potentially influence treatment outcomes following LTs. These factors included sex, age at LT, the type of LT, the time interval between the primary diagnosis and LT (≤12 months vs. >12 months), the median time interval between the primary diagnosis and LT, the median number of therapies administered (in addition to primary surgery) before LT, the median size of the treated tumour, the median Ki67 index of the primary tumour, the presence of autonomous glucocorticoid excess (yes vs. no), the location of the treated lesion, the number of lesions not treated with LT, and concurrent mitotane plasma concentration (maximum plasma level during LT ≤ 14 mg/L vs. >14 mg/L).

In a multivariable analysis employing the Cox proportional hazards model, we adjusted for all factors with a *p*-value less than 0.1 in the univariate analysis for tTTP, oPFS, and OS.

All reported *p*-values are two-sided, and statistical significance was considered when *p* < 0.05. Data analysis was conducted using SPSS version 29 (IBM SPSS Statistics).

## 4. Results

### 4.1. Patient Characteristics

The total cohort consisted of 66 patients with 132 individual lesions treated with LTs. Key patients’ characteristics are given in Table 1. All patients suffered from advanced ACC at the time they were treated with LT. None of these patients’ characteristics were significantly different between the three groups of LT.

### 4.2. Local Therapy Characteristics

Table 2 provides details of the LT modalities and the treated lesions. Of a total of 132 tumoural lesions, 84 were treated with imaging-guided ablation therapy (RFA, n = 64; MWA, n = 18; CA, n = 2), 40 with TA(C)E, and 8 with TARE. Lesions treated by TA(C)E were significantly larger in comparison to the other lesions. Among the patients for whom TA(C)E or TARE were applied, significantly higher tumour burden was recorded in comparison to patients in the imaging-guided ablation therapy group (LTA). Concomitant treatment with mitotane and mitotane plasma level were also significantly different between groups (see Table 2). 

### 4.3. Clinical Outcome According to Treatment Groups

Of 66 patients, 46 died due to progressive disease during follow-up. Median time of follow-up of surviving patients was 24.9 (14.4–48.3) months. 

The best objective response of the 132 lesions was complete response in 27 lesions (20.5 %), partial response in 27 lesions (20.5 %), and stable disease in 38 (28.8 %), leading to a disease control rate of 69.8%. Objective response was scattered among the different treatment groups, but complete response was only achieved by LTA (Table 3). In addition, we compared treatment efficacy depending on the location of the treated tumoural lesions. Overall, objective response rate was similarly distributed (*p* = 0.19). 

In 34 of the 132 lesions (25.7%), progression was documented at the first imaging after LT; overall, progress was diagnosed in 40 lesions during follow-up. tTTP was significantly different between groups. For the LTA group, the median tTTP was not reached, because only 17 lesions progressed during follow-up. However, the Kaplan–Meier analysis clearly indicates that tTTP was more than 22 months, whereas the median tTTP was 8.3 months for the TA(C)E group and 8.2 months in patients treated with TARE (*p* < 0.001; Figure 1). 

In univariable analysis, the following factors were associated with improved tTTP: modality of LT, median Ki67 index of the primary tumour, median time interval from primary diagnosis to LT, median number of therapies before LT, and mitotane plasma level during LT (Table 4). Using a multivariable model (with the LTA group as reference), time to local progression was significantly shorter in the TA(C)E and TARE group. Among the other variables, fewer therapies before LT, a higher mitotane plasma level and a longer time interval from primary diagnosis to LT remained significant (Table 4). 

In total, 97 cases with progression of lesions not treated with LTs were documented during follow-up and median overall PFS was 8.2 months. oPFS was significantly longer in the LTA group (23.4 months) compared to 8.3 months in the TA(C)E group and 1.6 months in the TARE group (*p* = 0.005). When adjusted in a multivariable analysis (with the LTA group as reference), oPFS was significantly shorter in the TA(C)E group (HR 2.24; 95% CI 1.18–4.02; *p* = 0.013) but not in the TARE group (HR 2.24; 95% CI 0.23–17.43; *p* = 0.44). Only the number of therapies before LT (*p* = 0.001) and the Ki67 index (*p* = 0.014) seemed to have an influence on oPFS (see Table 5).

At the last follow-up, 16 (33.3%) patients in the LTA group, 3 (20%) patients in the TA(C)E group, and 1 (12.5%) in the TARE group were still alive. Median overall survival in the LTA group was 33.3 months, for TA(C)E it was 14.8 months, and in the TARE group it was 18.9 months (*p* < 0.001). After multivariable analysis, overall survival in comparison to LTA was significantly shorter in the TA(C)E group (HR 3.74; 95% CI 1.99–7.01; *p* = 0.001) but not in comparison with the TARE group (HR 1.42; 95% CI 0.55–3.66; *p* = 0.47). Median Ki67 index >18% (*p* = 0.025), presence of glucocorticoid excess (*p* = 0.001), number of metastases >1 not treated with LT (*p* = 0.003), and median number of therapies before LT > 3 (*p* = 0.001) led to significantly shorter OS (see Table 6). 

### 4.4. Clinical Outcomes According to the Potential Intention of Treatment

We analysed tTTP in the three treatment groups according to curative or palliative treatment intention. In the 30 lesions treated with a “potentially curative approach”, complete response was achieved in 12 and only 3 progressed during follow-up (10 %), whereas this was the case in 41 of 102 lesions in “palliative approach” (40.2%). Accordingly, the median tTTP was not reached in the curative group vs. the median tTTP of 8.3 months in the palliative treatment group (*p* < 0.001). 

### 4.5. Adverse Events in Patients with Local Therapies

In the medical records, the documented adverse events associated with LT were mostly mild or moderate and typical for LT. Two episodes of bleeding (one grade 2 and one grade 3) following RFA of lung lesions were reported. In addition, three hepatic and two intestinal grade 1–2 adverse events were reported as infection following TA(C)E. Overall, the heterogeneity of documentation in medical records precluded more detailed assessment of adverse events.

## 5. Discussion

This retrospective analysis collected the largest series of patients with advanced ACC who underwent treatment with LTs. Our data indicate that LTs are of benefit for the majority of patients, in line with the findings of previous, smaller studies [31,41,44] and current guideline recommendations; these were mainly based on expert opinions [11,12]. The disease control rate was about 70%, and in 20% of lesions, complete response was achieved. 

Regarding the time to progression of the treated lesion (tTTP), the comparison of the different LT modalities suggests that treatment with LTA seemed to be more efficient than the other methods. The long median tTTP by LTA is remarkable. When comparing the three treatment groups, one has to acknowledge that the TARE group was very small, with just eight therapies in three patients. Other possible influencing factors could be the size of the treated lesion, which was significantly larger in the TA(C)E group, and the differences in mitotane therapy between the three groups. However, the size of the treated lesion did not statistically significantly influence tTTP. This was surprising but could be due to a relatively large number of lesions where size was incompletely recorded (n = 29). Apart from the LT modality, both the longer time interval between primary diagnosis and LT and the lower number of other therapies before LT were associated with statistically significantly longer tTTP in a multivariable analysis. This suggests that less aggressive disease was selected for LTs.

Of note, the longer tTTP observed for LTA may be biased by the selection of method, depending on the extent of disease and the clinical dynamics of disease evolution.

Finally, mitotane plasma levels higher than 14 mg/L remained significantly associated with better outcomes in this analysis, suggesting that concomitant mitotane treatment could be of benefit for these patients. Regarding oPFS and OS, LTA was more effective than TA(C)E and TARE, but these differences were only significant in comparison to the TA(C)E group. The reported toxicities were few and moderate, similar to the previously published studies on LTs in ACC [31,44], but under-reporting is likely in these retrospective studies. However, it is probable that at least severe adverse events would have been reported in the medical records. The fact that only one grade 3 event, namely a pulmonary bleeding, occurred does not prove that LTs are always safe, but suggests that the likelihood of major complications is probably low. 

Within the LTA group, we could not determine whether one of the procedures is more effective than the others, because the number of patients treated with MWA and CA were too small. However, MWA might offer advantages due to higher constant intra-tumoural temperatures, generating heat in larger volume of tissue, and leading to larger ablation zones, faster ablation times, and the ability to use multiple probes to treat multiple lesions simultaneously. Another advantage of MWA is that fewer applicators are needed, and ablative margins are easier to obtain than with RFA [52]. Therefore, MWA is usually considered as the technique of choice for larger tumours or when the tumour is close to large vessels, independent of its size [53,54]. Some studies in other tumour types have suggested that CA is superior to other LTAs and leads to quicker recoveries compared to heat-based ablation therapies [55]. One important benefit of CA is that the ablation zone is easily visualised by US, CT, and magnetic resonance imaging, thus allowing relatively more precise monitoring of the ablation zone than is possible with many heat-based systems [52]. In the end, multiple factors should be taken into account when deciding on LTs in ACC. Besides organ-specific considerations and tumour location (especially proximity to vulnerable structures and to blood vessels), local expertise seems to be of major relevance. 

It is a main difference between our study and recently published studies in ACC that we could not find a significant association of the size of the treated lesion with clinical outcome. In contrast, Li et al. suggested that microwave ablation is particularly suitable for tumours smaller than 5 cm [46] and Wood et al. suggested that RFA is most effective in tumours < 5 cm [42]. In addition, three other studies reported better clinical outcomes in lesions with a diameter ≤2 cm [43] or <3 cm [31,41], respectively. Although we cannot fully elucidate in our retrospective setting why size did not influence the outcome in our series in a relevant manner, it is tempting to speculate that patient selection in our more recent study was influenced by the results published in these earlier studies. This is especially the case in the LTA group, where the median size of the treated lesions was only 2 cm and 80% were smaller than 3 cm. As pointed out before, the aggression of the tumour might be a potential factor influencing the success of LTAs. Several years ago, Mauda-Havakuk et al. reported a longer time from diagnosis to first thermal ablation as a potential predictor of prolonged survival in a small series of 12 patients [44]. However, most other studies did not even analyse this factor. 

The critical aspect of applying local therapies to ACC patients likely lies in selecting the most appropriate method for each individual patient. Despite the paucity of published evidence, our study further points towards a benefit of LTs in ACC, particularly in patients with oligo-metastatic, slowly progressing disease. In this subgroup, some patients experienced disease control for more than 3 years, and the start of cytotoxic therapy could, therefore, be significantly delayed. The fact that patients with a mitotane level > 14 mg/L at the time of the LT had a longer tTTP than patients without mitotane treatment or lower plasma level does not prove that co-treatment with mitotane is of benefit. However, our study strongly supports the proposal by Baudin and colleagues to combine LTAs with mitotane [56].

Our study possesses evident limitations, including its retrospective nature, the relatively small number of patients, and the absence of a control group. Given the rarity of the disease, obtaining a larger number of treated lesions is challenging. One of the major limitations of this study is the uncertainty regarding adverse events. Although the reported toxicity appears to be limited, it is important to recognise that our study may tend to underestimate negative effects due to its retrospective nature. However, data from other studies investigating LTs in advanced ACC, but also larger studies in other tumour entities, suggest that most local treatment modalities have acceptable toxicity levels [42,43,44,57,58,59]. Another limitation concerns the diversity in LT modalities and varying group sizes. It is important to acknowledge that the decision for LTs was made by local treating physicians and was not based on predefined criteria. Additionally, the application of LTs was not standardised, and the same holds true for co-treatment with mitotane. 

## 6. Conclusions

In conclusion, our study demonstrates that LTs are associated with beneficial effects on clinical outcomes in selected patients with advanced ACC. These results are in line with previous, smaller studies and reinforce the idea that LTs are underused and should be considered as treatment options in patients with advanced ACC, not just as palliative therapy. Our study particularly suggests that local thermal ablation therapies in patients with less-aggressive tumour behaviour might be an effective therapeutic approach. In addition, we provide some evidence that co-treatment with mitotane could be of added value. However, further studies need to confirm these predictive factors. 

## Figures and Tables

**Figure 1 cancers-16-00706-f001:**
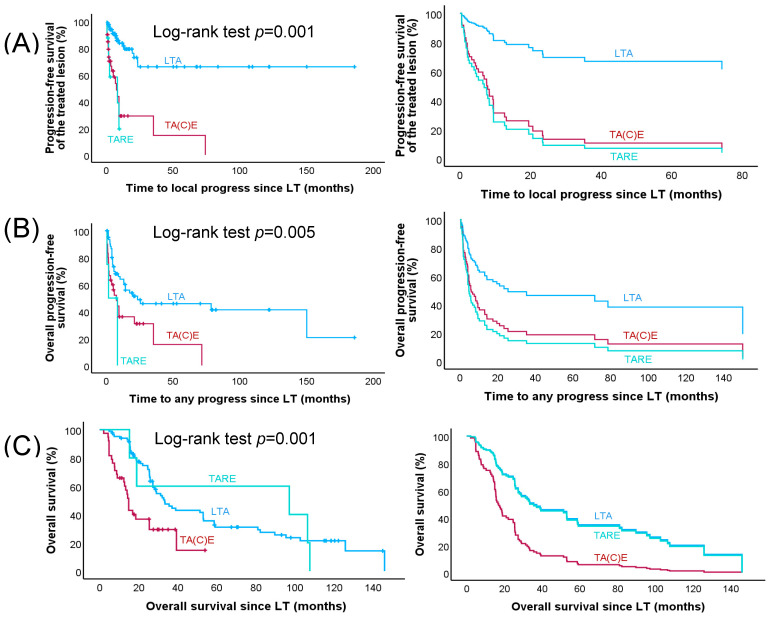
Time to local progress, time to any progress, and overall survival since LT in all patients with advanced ACC. Kaplan–Meier (left panel) and multivariable Cox regression (right panel) survival curves (**A**) for time to progress of the treated lesion (tTTP), (**B**) overall progression-free survival (oPFS), and (**C**) overall survival (OS). LTA (image-guided local ablation therapy) (n = 84), TA(C)E (transarterial (chemo)embolization) (n = 40), and TARE (transarterial radioembolization) (n = 8)) are included in the Kaplan Meier curves, whereas—due to missing information on tumour size—only 5 instead of 8 lesions treated with TARE were included in the multivariable Cox regression analyses.

**Table 1 cancers-16-00706-t001:** Baseline characteristics of the patients.

	LTA n = 48	TA(C)E n = 15	TARE n = 3	*p*
Median age at primary diagnosis—years (IQR)	47.4 (31.7–57.9)	41.4 (29.1–54.4)	38.4 (29.3–38.4)	0.72
Median age at LT—years (IQR)	49.4 (31.2–61.5)	48.6 (32.3–61.4)	40.9 (29.8–40.9)	0.98
Sex—n (%)				0.62
male	14 (29.2)	5 (33.3)	0	
female	34 (70.8)	10 (66.7)	3 (100)	
Glucocorticoid excess—n (%)	27 (56.3)	4 (26.7)	1 (33.3)	0.14
ENSAT stage at primary diagnosis—n (%)				0.44
1	6 (12.5)	2 (13.3)	0	
2	20 (41.7)	5 (33.3)	0	
3	16 (33.3)	4 (26.7)	3 (100)	
4	6 (12.5)	4 (26.7)	0	
Resection status at primary diagnosis—n (%)				0.47
0	32 (66.7)	11 (73.3)	1 (33.3)	
1	4 (8.3)	0	0	
2	2 (4.2)	2 (13.3)	0	
X	8 (16.6)	1 (6.7)	2 (66.7)	
Missing data	2 (4.2)	1 (6.7)		
Median Ki67 index of the primary tumour—% (IQR)	15 (10–25)	20 (14–27.5)	17.5 (2–17.5)	0.18

LTA—image-guided local ablation therapy; TA(C)E—transarterial (chemo)embolisation; TARE—transarterial radioembolisation; n—number; IQR—interquartile range; LT—local therapy.

**Table 2 cancers-16-00706-t002:** Baseline characteristics of LT.

	LTA n = 84	TA(C)E n = 40	TARE n = 8	*p*
Location of treated lesion—n (%)				0.081
Local recurrence	2 (2.4)	2 (5)	0	
Lung	19 (22.6)	1 (2.5)	1 (12.5)	
Bone	5 (6)	0	0	
Liver	49 (58.3)	30 (75)	7 (87.5)	
Lymph node	1 (1.2)	0	0	
Other soft tissue	8 (9.5)	7 (17.5)	0	
Median size of lesion—mm (IQR)	20.5 (12–30)	61.5 (24.3–108)	49.0 (42–77.5)	0.001
Median needle length for RFA—cm (IQR)	3 (2–3.5)			
Median duration RFA—min (IQR)	11 (10–13.5)			

RFA modality—n (%)				
Percutaneous	63 (98.4)			
Laparoscopic	0			
Open surgery	1 (1.6)			
Median dosage microwave ablation—Watt (IQR)	100 (100–110)			
Embolic particles/drug used for TA(C)E—n (%)				
Mitomycin		1 (2.8)		
Irinotecan		5 (14.3)		
Doxorubicin		2 (5.7)		
Epirubicin + cisplatin		4 (11.4)		
Gemcitabine + cisplatin		2 (5.7)		
Irinotecan + cisplatin		4 (11.4)		
Microspheres/particles		10 (28.6)		
Lipiodol only		4 (11.4)		
Lipiodol + microspheres		3 (8.9)		
Use of lipiodol yes—n (%)		22 (55)		
Median number of therapies (in addition to primary surgery) before LT—n (IQR)	2 (1–4)	3 (3–5)	5 (2–6)	0.055
Median time interval between primary diagnosis and start of LT—months (IQR)	51 (17.9–94.9)	39.8 (20.3–57.9)	41.9 (11.3–55.7)	0.52
Median number of lesions not treated with LT—n (IQR)	1 (0–1)	3 (1–4)	3 (3–3)	0.001
Concomitant mitotane during LT—n (%)	49 (58.3)	19 (47.5)	0	0.003
Mitotane plasma level > 14 mg/L during LT				
Yes (%)	20 (23.8)	8 (20)	NA	0.05
Median time to first imaging—months (IQR)	2.4 (1.1–2.4)	1.3 (0.9–3.9)	2.3 (1.2–2.3)	0.12
Median time to second imaging—months (IQR)	5.7 (3.2–10.8)	4.7 (3.6–6.8)	3.5 (2.5–9.4)	0.26

LTA—image-guided local ablation therapy; TA(C)E—transarterial (chemo)embolisation; TARE—transarterial radioembolisation; n—number; IQR—interquartile range; LT—local therapy; NA—not applicable; mm—millimetre; cm—centimetre; min—minute.

**Table 3 cancers-16-00706-t003:** Objective response according to the different treatment groups.

	Number of Lesions n (%)	Complete Response n (%)	Partial Response n (%)	Stable Disease n (%)	Progressive Disease n (%)
LTA	84	27 (32.1)	13 (15.5)	27 (32.1)	17 (20.2)
TA(C)E	40	0	12 (30)	10 (25)	18 (45)
TARE	8	0	2 (25)	1 (12.5)	5 (62.5)
Location of treated lesion					
Local recurrence	4	1 (25)	1 (25)	1 (25)	1 (25)
Lung	21	3 (14.3)	5 (23.8)	9 (42.9)	4 (19)
Bone	5	0	1 (20)	1 (20)	3 (60)
Liver	86	18 (20.9)	13 (15.1)	25 (29.2)	30 (34.8)
Lymph node	1	0	0	1 (100)	0
Other soft tissue	15	5 (33.3)	7 (46.7)	1 (6.7)	2 (13.3)

LTA—image-guided local ablation therapy; TA(C)E—transarterial (chemo)embolisation; TARE—transarterial radioembolisation; n—number.

**Table 4 cancers-16-00706-t004:** Predictive factors for tTTP.

	n	Median tTTP (Months)	Univariable Analysis			Multivariable Analysis		
			HR	95% CI	*p*	HR	95% CI	*p*
Treatment group								
1. LTA	84	Not reached	1			1		
2. TA(C)E	40	8.3	5.59	2.92–10.74	0.001	4.82	2.04–11.36	0.001
3. TARE	8	8.2	6.65	1.89–23.48	0.003	6.16	1.48–25.56	0.012
Age at start LT ^1^								
≤48	63	Not reached	1					
>48	69	23.5	1.13	0.61–2.10	0.69			
Sex								
female	100	74.2	1					
male	32	20.6	1.13	0.56–2.25	0.74			
Ki67 of the primary tumour ^1^								
≤18%	64	74.2	1			1		
>18%	53	9.3	2.63	1.39–4.96	0.003	1.97	0.74–5.41	0.19
Glucocorticoid excess								
no	74	23.5	1					
yes	58	Not reached	0.63	0.33–1.20	0.16			
Location								
1. Liver	86	20.6	1					
2. Pulmonary	21	Not reached	0.42	0.16–1.10	0.08			
3. LR	4	5.2	1.10	0.26–4.62	0.89			
4. Bone	5	7.5	0.97	0.23–4.10	0.96			
5. LN	1	6.5	NA	NA	NA			
6. Soft tissue	15	Not reached	0.32	0.075–1.33	0.12			
Size of the treated lesion ^1^								
≤24 mm	52	13.9	1					
>24 mm	51	19.5	1.82	0.82–4.04	0.14			
Number of metastases without LT ^1^								
≤1	50	Not reached	1					
>1	28	12.6	1.48	0.65–3.40	0.35			
Time interval primary diagnosis—LT								
≤12 months	17	74.2	1					
>12 months	115	35.3	1.15	0.45–2.94	0.77			
Time interval primary diagnosis—LT ^1^								
≤47 months	66	19.5	1			1		
>47 months	66	Not reached	0.48	0.25–0.89	0.020	0.21	0.07–0.64	0.006
Number of therapies before LT ^1^								
≤3	87	Not reached	1			1		
>3	45	12.6	2.24	1.26–4.12	0.010	6.67	2.49–17.88	0.001
Mitotane plasma level during LT								
≤14 mg/L	56	13.1	1			1		
>14 mg/L	28	Not reached	0.30	0.12–0.74	0.009	0.36	0.14–0.93	0.035

Due to missing data on tumour size, three lesions treated with TARE were excluded from the Cox regression analyses. Only factors that showed at least a trend in the univariate analysis with *p* < 0.1 were further investigated by multivariable analysis. LR—local recurrence; LN—lymph node; HR—hazard ratio; n—number; mm—millimetre; mg/L—milligram per litre. ^1^ These parameters were categorised by splitting the group on the median. Cox regression analyses included 129 instead of 132 lesions due to the exclusion of three lesions with TARE.

**Table 5 cancers-16-00706-t005:** Predictive factors for oPFS.

	n	Median oPFS (Months)	Univariable Analysis			Multivariable Analysis		
			HR	95% CI	*p*	HR	95% CI	*p*
Treatment group								
1. LTA	63	23.4	1			1		
2. TA(C)E	30	8.3	2.19	1.25–3.87	0.007	2.18	1.18–4.02	0.013
3. TARE	4	1.6	2.72	0.37–20.19	0.33	2.24	0.23–17.43	0.44
Age at start LT ^1^								
≤48	46	71.6	1					
>48	51	13.1	1.54	0.87–2.71	0.14			
Sex								
female	74	17.5	1					
male	23	9.5	1.29	0.68–2.46	0.43			
Ki67 of the primary tumour ^1^								
≤18%	49	23.4	1			1		
>18%	38	4.9	2.34	1.32–4.14	0.004	2.42	1.97–4.88	0.014
Glucocorticoid excess								
no	57	10.3	1					
yes	40	78.6	0.64	0.36–1.13	0.12			
Location								
1. Liver	62	8.3	1					
2. Pulmonary	13	150.3	0.28	0.097–0.79	0.016			
3. LR	4	5.2	0.75	0.18–3.11	0.69			
4. Bone	4	1.7	1.49	0.45–4.94	0.51			
5. LN	0	NA	NA	NA	NA			
6. Soft tissue	14	19.6	0.63	0.28–1.42	0.27			
Size of the treated lesion ^1^								
≤24 mm	44	14.3	1					
>24 mm	34	17.5	1.15	0.61–2.15	0.68			
Number of metastases without LT ^1^								
≤1	43	19.6	1			1		
>1	24	8.3	2.28	1.16–4.49	0.017	2.19	0.94–5.06	0.068
Time interval primary diagnosis—LT								
≤12 months	14	4.1	1					
>12 months	83	17.5	0.71	0.35–1.43	0.34			
Time interval primary diagnosis—LT ^1^						1		
≤47 months	49	6.7	1			0.56	0.29–1.10	0.088
>47 months	48	25.5	0.54	0.32–0.94	0.030			
Number of therapies before LT ^1^								
≤3	69	21.0	1			1		
>3	28	9.5	2.01	1.14–3.57	0.016	3.62	1.87–7.01	0.001
Mitotane plasma level during LT								
≤14 mg/L	38	9.5	1					
>14 mg/L	23	8.3	0.89	0.47–1.67	0.72			

Only factors that showed at least a trend in the univariate analysis with *p* < 0.1 were further investigated by multivariable analysis. LR—local recurrence; LN—lymph node; HR—hazard ratio; n—number; mm—millimetre; mg/L—milligram per litre. ^1^ These parameters were categorised by splitting the group on the median. Cox regression analyses included 94 instead of 97 lesions due to the exclusion of three lesions with TARE.

**Table 6 cancers-16-00706-t006:** Predictive factors for OS.

	n	Median OS (Months)	Univariable Analysis			Multivariable Analysis		
			HR	95% CI	*p*	HR	95% CI	*p*
Treatment group								
1. LTA	84	33.3	1			1		
2. TA(C)E	40	14.8	2.70	1.65–4.43	0.001	3.74	1.99–7.01	0.001
3. TARE	8	18.9	1.03	0.41–2.58	0.95	1.42	0.55–3.66	0.47
Age at start LT ^1^								
≤48	63	28.7	1					
>48	69	27.9	1.21	0.78–1.85	0.39			
Sex								
female	100	27.2	1					
male	32	32.7	0.91	0.56–1.46	0.68			
Ki67 of the primary tumour ^1^								
≤18%	64	59.1	1			1		
>18%	53	15.9	3.40	2.08–5.56	0.001	2.67	1.13–6.32	0.025
Glucocorticoid excess								
no	74	37.0	1			1		
yes	58	22.0	1.44	0.94–2.19	0.092	4.08	1.76–9.43	0.001
Location								
1. Liver	86	30.4	1					
2. Pulmonary	21	Not reached	0.41	0.22–0.78	0.007			
3. LR	4	4.5	0.55	0.14–2.26	0.41			
4. Bone	5	3.6	0.97	0.39–2.40	0.94			
5. LN	1	6.5	NA	NA	NA			
6. Soft tissue	15	20.4	0.18	0.044–0.73	0.017			
Size of the treated lesion ^1^								
≤24 mm	52	27.2	1					
>24 mm	51	35.5	1.10	0.67–1.75	0.74			
Number of metastases without LT ^1^								
≤1	50	32.8	1			1		
>1	28	17.9	3.57	1.89–6.88	0.001	3.39	1.52–7.55	0.003
Time interval primary diagnosis—LT								
≤12 months	17	25.8	1					
>12 months	115	32.0	0.98	0.54–1.81	0.94			
Time interval primary diagnosis—LT ^1^								
≤47 months	66	25.4	1			1	0.56–1.86	
>47 months	66	53.0	0.54	0.35–0.83	0.005	1.03		0.94
Number of therapies before LT ^1^								
≤3	87	31.2	1			1	2.36–12.65	
>3	45	24.6	2.10	1.35–3.18	0.001	5.47		0.001
Mitotane plasma level during LT								
≤14 mg/L	56	25.5	1					
>14 mg/L	28	25.8	0.70	0.38–1.26	0.23			

Factors that showed at least a trend in the univariate analysis with *p* < 0.1 were further investigated by multivariable analysis. LR—local recurrence; LN—lymph node; HR—hazard ratio; n—number; mm—millimetre; mg/L—milligram per litre. ^1^ These parameters were categorised by splitting the group on the median. Cox regression analyses included 129 instead of 132 lesions due to the exclusion of three lesions with TARE.

## Data Availability

ACC is an ultrarare disease and patient privacy is important. However, we are committed to sharing data with all qualified external researchers. Requests have to be sent to the corresponding authors. All data provided are anonymised to respect the privacy of the patients.

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
