# Peer review of "The Value of Local Therapies in Advanced Adrenocortical Carcinoma"

_cancers, 2024, doi:10.3390/cancers16040706_

Round 1
Reviewer 1 Report
Comments and Suggestions for Authors
In this manuscript, Kimpel et al. have retrospectively investigated the potential value of local therapies in advanced adrenocortical carcinomas, as until now only minor studies on this topic have been published. The authors conclude that local therapies should continue to be recommended, as their study demonstrates that, in particular, local thermal ablation therapies can be associated with beneficial effects in patients with less aggressive advanced adrenocortical carcinomas.
The study clearly describes the benefits of the different local therapies, but what about complications by the therapies, such as, for example, bleedings or post ablation syndrome? You just write that the toxicity is low but without any details. Maybe this can be discussed further.
Otherwise, the manuscript is very well written and nicely structured. Therefore, I only have some minor comments:
11) Lines 27 and 92: In the sentence “….clear recommendations which patients….”, I would write “…clear recommendations regarding which patients…”
22) For the LTA group, it is not possible to write the median time to progression of the treated lesion (tTTP) because it was not reached yet, but would it at least be possible to say higher than a specific number of months?
33) Line 84. Remove “together”.
44) Line 145: You write that the maximum induced heat was 140 Watt. Power but not temperature is measured in Watt. For how long was this power applied and what was the maximum achieved temperature?
55) Table 3: There is a mistake in line 5 of the table, where it says “Localiz Localisation”. There is also a problem with the lines that have been partially shifted.
66) Line 305: “recordes” should be “records”
77) Lines 321-22: “…and the different mitotane therapy…” should be changed to for example “…the differences in mitotane therapy…”
88) Line 330: To “plasma mitotane” I would add levels or concentrations.
Author Response
Reviewer 1
In this manuscript, Kimpel et al. have retrospectively investigated the potential value of local therapies in advanced adrenocortical carcinomas, as until now only minor studies on this topic have been published. The authors conclude that local therapies should continue to be recommended, as their study demonstrates that, in particular, local thermal ablation therapies can be associated with beneficial effects in patients with less aggressive advanced adrenocortical carcinomas.
The study clearly describes the benefits of the different local therapies, but what about complications by the therapies, such as, for example, bleedings or post ablation syndrome? You just write that the toxicity is low but without any details. Maybe this can be discussed further.
Otherwise, the manuscript is very well written and nicely structured. Therefore, I only have some minor comments:
OUR RESPONSE: We fully appreciate the overall positive comment of the reviewer and would have been happy if we would be able to report the rate of complications more detailed and especially with more confidence. However, due to the retrospective nature of the study, adverse events were only documented in a small group of patients. We now describe the reported complications more detailed at line 311ff. Particularly we found 2 bleedings following RFA of lung lesions, and 5 infections, including 3 hepatic and 2 intestinal infections, following TA(C)E. In addition, we refer to these adverse events in the discussion (line 341ff). Finally, we now mention this aspect as a major weakness of our study in line 388ff.
1) Lines 27 and 92: In the sentence “….clear recommendations which patients….”, I would write “…clear recommendations regarding which patients…”
OUR RESPONSE: We followed the suggestion of the reviewer and corrected the sentence.
2) For the LTA group, it is not possible to write the median time to progression of the treated lesion (tTTP) because it was not reached yet, but would it at least be possible to say higher than a specific number of months?
OUR RESPONSE: Following the suggestion of the reviewer, we reported now that the median tTTP was at least 22 months and emphasize that only 17 lesions progressed at all during follow-up (see line 243ff).
3) Line 84. Remove “together”.
OUR RESPONSE: We have now corrected it.
4) Line 145: You write that the maximum induced heat was 140 Watt. Power but not temperature is measured in Watt. For how long was this power applied and what was the maximum achieved temperature?
OUR RESPONSE: We thank the reviewer for pointing to this error that we have now corrected and report “power” instead of temperature, which was usually not documented. The maximum duration of the procedure was six minutes. This information is now given in lines 149-150.
5) Table 3: There is a mistake in line 5 of the table, where it says “Localiz Localisation”. There is also a problem with the lines that have been partially shifted.
OUR RESPONSE: Thank you to point this out. These errors are now corrected in the revised manuscript.
6) Line 305: “recordes” should be “records”
OUR RESPONSE: We have now corrected this typo.
7) Lines 321-22: “…and the different mitotane therapy…” should be changed to for example “…the differences in mitotane therapy…”
OUR RESPONSE: We have now corrected this.
8) Line 330: To “plasma mitotane” I would add levels or concentrations.
OUR RESPONSE: We included the plasma mitotane level in this sentence.
Reviewer 2 Report
Comments and Suggestions for Authors
Kimpel and coworkers investigated retrospectively the effects of different types of local therapies in 66 patients with 132 lesions of advanced adrenocortical carcinoma, by studying in particular the effects of thermal ablation therapies(LTA ,n.84), Transarterial(chemo)embolization(TA(C)E, n.40), transarterial radioembolization(TARE,n.8). A possible complete remission was obtained in 21% of treated tumoral lesions with the best long-term remission in patients treated with LTA . They conclude that particularly LTA may be an effective therapeutic approach in patients with a rather less aggressive behaviour and that the co-treatment with mitotane may improve the results.
Comments
This work follows the lines of previous research by the same authors on this topic, whose published papers are abundantly self-cited in the bibliography, whereas some review by others have been ignored (For example: Alyateem G, Nilubol N. Current status and future targeted therapy in adrenocortical cancer. Front Endocrinol 2021). In particular, on this topic the authors published also a review(ref.18 of this paper). I am amazed that the authors published a review while the paper with original data on this topic was stil in peer-review . Concerning this, strangely, the citation 18 of their paper : " Kimpel OD,; U.; Altieri, A; Fuss,CT; Polat,B; Kickuth, R; Kroiss, M; Fassnacht, M. Current evidence on local therapies in advanced adrenocortical carcinoma. Hormone and Metabolic Research 2023", is incorrect and incomplete, because the review has been published in 2024 and the correct citation is: " Kimpel, O; Dischinger,U; Altieri,A; Fuss,CT; Polat, B; Kickuth, R; Kroiss,M; Fassnacht, M. Current evidence on local therapies in advanced adrenocortical carcinoma. Horm. Metab Res 2024, Jan ; 56(1):91-98. doi: 10.1055/a-2209-6022. Epub 2024 Jan 3". I think that the authors should clarify these aspects before continuing the peer review of their paper.
Author Response
Reviewer 2
Kimpel and coworkers investigated retrospectively the effects of different types of local therapies in 66 patients with 132 lesions of advanced adrenocortical carcinoma, by studying in particular the effects of thermal ablation therapies(LTA ,n.84), Transarterial(chemo)embolization(TA(C)E, n.40), transarterial radioembolization(TARE,n.8). A possible complete remission was obtained in 21% of treated tumoral lesions with the best long-term remission in patients treated with LTA . They conclude that particularly LTA may be an effective therapeutic approach in patients with a rather less aggressive behaviour and that the co-treatment with mitotane may improve the results.
Comments
This work follows the lines of previous research by the same authors on this topic, whose published papers are abundantly self-cited in the bibliography, whereas some review by others have been ignored (For example: Alyateem G, Nilubol N. Current status and future targeted therapy in adrenocortical cancer. Front Endocrinol 2021). In particular, on this topic the authors published also a review(ref.18 of this paper). I am amazed that the authors published a review while the paper with original data on this topic was stil in peer-review . Concerning this, strangely, the citation 18 of their paper : " Kimpel OD,; U.; Altieri, A; Fuss,CT; Polat,B; Kickuth, R; Kroiss, M; Fassnacht, M. Current evidence on local therapies in advanced adrenocortical carcinoma. Hormone and Metabolic Research 2023", is incorrect and incomplete, because the review has been published in 2024 and the correct citation is: " Kimpel, O; Dischinger,U; Altieri,A; Fuss,CT; Polat, B; Kickuth, R; Kroiss,M; Fassnacht, M. Current evidence on local therapies in advanced adrenocortical carcinoma. Horm. Metab Res 2024, Jan ; 56(1):91-98. doi: 10.1055/a-2209-6022. Epub 2024 Jan 3". I think that the authors should clarify these aspects before continuing the peer review of their paper.
OUR RESPONSE: We are happy to clarify this issue, because we are convinced that the expert's view is not entirely correct. It is true that we have cited many paper with the participation of co-authors from our lead centre. However, it should be borne in mind that the University Hospital of Würzburg, one of the world's largest centres for ACC, has been publishing on this topic for more than 20 years. In this context, 13 out of 48 references (including several important papers such as guidelines that "must" be cited) do not appear to us to be excessively self-citing. However, we have now added 11 more references from other groups (including the review suggested by the reviewer). Regarding the cited review "Kimpel et al Horm Metab Res", the reviewer is correct that this manuscript was indeed officially published on 3 January 2024. As the acceptance was reported many weeks earlier, we assumed publication in 2023 and apologise for not updating this reference correctly. This has of course been corrected in the current version. However, the assumption that this review contains data from this original study is simply incorrect. The report was written many months before this study was analysed. The fact that the review and publication process took so long was not our responsibility.
Round 2
Reviewer 2 Report
Comments and Suggestions for Authors
I have no comments and suggestions for authors